# Comparative RNA-Seq analyses of *Drosophila* plasmatocytes reveal gene specific signatures in response to clean injury and septic injury

**Elodie Ramond**[ID]*[◉], **Jan Paul Dudzic**[ID]*[◉], **Bruno Lemaitre**[ID]*

Global Health Institute, School of Life Science, École Polytechnique Fédérale de Lausanne (EPFL), Lausanne, Switzerland

◉ These authors contributed equally to this work.
* elodie.ramond@inserm.fr (ER); jandudzic@uvic.ca (JPD); bruno.lemaitre@epfl.ch (BL)

**Data Availability Statement:** Data are available within the manuscript and its Supporting Information files. The raw data of the RNA-seq study are deposited at the NCBI sequence read

## Abstract

*Drosophila melanogaster*'s blood cells (hemocytes) play essential roles in wound healing and are involved in clearing microbial infections. Here, we report the transcriptional changes of larval plasmatocytes after clean injury or infection with the Gram-negative bacterium *Escherichia coli* or the Gram-positive bacterium *Staphylococcus aureus* compared to hemocytes recovered from unchallenged larvae via RNA-Sequencing. This study reveals 676 differentially expressed genes (DEGs) in hemocytes from clean injury samples compared to unchallenged samples, and 235 and 184 DEGs in *E. coli* and *S. aureus* samples respectively compared to clean injury samples. The clean injury samples showed enriched DEGs for immunity, clotting, cytoskeleton, cell migration, hemocyte differentiation, and indicated a metabolic reprogramming to aerobic glycolysis, a well-defined metabolic adaptation observed in mammalian macrophages. Microbial infections trigger significant transcription of immune genes, with significant differences between the *E. coli* and *S. aureus* samples suggesting that hemocytes have the ability to engage various programs upon infection. Collectively, our data bring new insights on *Drosophila* hemocyte function and open the route to post-genomic functional analysis of the cellular immune response.

## Introduction

*Drosophila* blood cells, also called hemocytes, contribute to the cellular immune response by engulfing bacteria, combatting parasites and secreting antimicrobial and clotting factors. They also participate in regulating the immune response by secreting cytokines such as the JAK-STAT ligands Unpaired [1] or the Toll ligand Spätzle [2–4]. Hemocytes are also involved in wound healing notably through the engulfment of apoptotic cells and cellular debris, the stimulation of stem cell proliferation, and deposition of extracellular matrix [5–8]. Furthermore, hemocytes produce enzymes essential to the melanization reaction [9,10]. Recent evidence shows that *Drosophila* blood cells contribute not only to immunity and wound healing, but are also central to host metabolism [11–14]. That an excessive number of hemocytes can be detrimental to flies raised on a poor diet shows that hemocyte number must be tightly regulated

archive (SRA) with the accession number
PRJNA638422.

**Funding:** ER, JPD and BL were supported by the
SNF (Swiss National Science Foundations) sinergia
grant CRSII5 186397.

**Competing interests:** The authors have declared
that no competing interests exist.

[15]. Thus, there is a current effort to better characterize the role of hemocytes during the life cycle of flies.

Hematopoiesis occurs in several waves throughout the *Drosophila* life cycle. The first phase of hematopoiesis establishes a pool of hemocytes from the embryonic head mesoderm. These cells contribute to embryonic development by phagocytosing apoptotic cells, and through the deposition of extracellular matrix [8]. These embryonic derived hemocytes persist in larvae, where they are subjected to several rounds of division reaching about 6000 hemocytes at the end of the third instar larval stage [16]. Peripheral larval hemocytes are found either (i) in circulation in the hemolymph or (ii) in sessile patches [17–23]. Sessile hemocytes are attached to the internal surface of the larval body wall, forming patches, some of which are closely associated with secretory cells called oenocytes, as well as the endings of peripheral neurons [22,24]. Hemocytes are continuously exchanged between sessile patches and circulation [25,26]. The function of sessile hemocyte patches is not yet established but it has been proposed that they form a diffuse hematopoietic organ [22,27,28]. Larvae also possess a special hematopoietic organ, the lymph gland, that functions as a reservoir releasing hemocytes at the pupal stage or after parasitic infection. Both lymph gland and embryonic derived hemocyte populations contribute to the pool of adult hemocytes that will ultimately decline upon ageing. Whether active hematopoiesis occurs in adults is still debated [29,30].

Most studies on the cellular immune response focus on third instar larval hemocytes as both sessile and circulating hemocytes can easily be collected and FACS sorted. *Drosophila* larvae have two types of hemocytes in the unchallenged state: plasmatocytes, which are macrophage-like, and crystal cells, rounded hemocytes which contain crystals of prophenoloxidases, the zymogen form of phenoloxidases that catalyzes the melanization reaction against parasites or septic injury [9,17,31]. A third type of hemocytes, the lamellocytes, is restricted to the larval stage and originates either from progenitors in the lymph gland or in periphery by transdifferentiation of plasmatocytes or circulating progenitors [32–34]. These cells differentiate upon parasitoid wasp infestation and contribute to the encapsulation and melanization of larger parasites. At the larval stage, plasmatocytes represent the most abundant fraction of *Drosophila* blood cells (i.e. 90–95%) [35] and express several markers such as the clotting factor Hemolectin (Hml), or the phagocytic receptors Nimrod C1 (NimC1 or P1) or Eater [18]. The other 5–10% larval hemocytes are Lozenge (Lz) positive crystal cells [17]. Only rarely can lamellocytes be observed in the unchallenged larvae as these cells are induced upon wasp infestation or injury [32].

Until recently, there have been surprisingly few studies analyzing the hemocyte transcriptome, possibly due to difficulties in collect enough materials. The most comprehensive genome wide analysis was a characterization of whole larval hemocyte populations by Irving et al. in 2005, using an Affymetrix based oligonucleotide array [2]. Of the 13,000 genes (total number of genes > 17,500) represented in this microarray, they were able to identify 2500 with significantly enriched expression in hemocytes, notably genes encoding integrins, peptidoglycan recognition proteins (PGRPs), scavenger receptors, lectins, cell adhesion molecules and serine proteases. Interestingly, several single cell transcriptomic analyses have revealed the degree of heterogeneity of *Drosophila* hemocyte populations, but they did not characterize the full repertoire of genes expressed in hemocytes [36–39].

To better characterize the transcriptome of hemocytes, we have performed an RNAseq transcriptome analysis of FACS sorted Hml positive cells. The transcriptome of Hml positive (Hml+) plasmatocytes was determined in an unchallenged condition and 45 minutes following clean or sceptic injury with *Staphylococcus aureus* or *Escherichia coli*. Comparative transcriptomics allowed us to identify a set of genes specific to plasmatocytes in unchallenged or challenged condition, revealing the various contributions of these cells to host defense, wound healing and metabolism.

## Results

### Study design

We performed RNA sequencing of mRNA to analyze the global gene expression profile changes of *Drosophila* hemocytes from third instar larvae either unchallenged or collected 45 minutes after clean injury or septic injury with a needle dipped in concentrated bacterial pellets of *Staphylococcus aureus* or *Escherichia coli*. To isolate the plasmatocytes from other unwanted cells of the hemolymph preparation, we used the *Hml*$^{\Delta}$.*Ds-Red.nls* fluorescent marker, which is specifically expressed in most plasmatocytes, and to a lesser extent in newly differentiated crystal cells [28,40,41]. We extracted hemolymph from wandering third instar L3 larvae by bleeding them onto a glass slide and subjected the collected hemolymph to fluorescence activated cell sorting (FACS) to isolate the *Hml*+ hemocyte population. The collected hemocytes thus correspond to circulating hemocytes. Flow cytometer scatter-plot outputs were analyzed to delineate the hemocyte population based on the nucleic red-fluorescent signal, and total RNA extraction was performed on the isolated hemocytes (**Fig 1**). We collected approximately 20,000 to 40,000 larval Hml-positive plasmatocytes for each treatment. Three independent extractions were performed for each tested condition. As in Irving et al, 2005, we used unchallenged whole larvae as an external control to identify genes that were specifically enriched in plasmatocytes compared to the whole animal. RNA-Seq libraries were then constructed and sequenced using Illumina HiSeq, and we performed differential gene expression analysis between all sample groups. We obtained 62,320,223 reads from RNA samples extracted from whole larvae (L3), 42,284,187 reads from hemocytes of unchallenged larvae (UC), 42,519,937 reads from RNA extracted from hemocytes of clean-injured larvae (CI), 69,143,536 reads from RNA extracted from hemocytes of *E. coli* infected larvae (Ec) and 42,758,456 reads from RNA extracted from hemocytes of *S. aureus* infected larvae (Sa) (sum of triplicates). The total number of mapped reads per single library ranged from 5.30 to 23.17 million reads, with coverage ranging from 59.42% to 86.14% (**Fig 2A**). Genes with more than 5 reads per 1 million reads are listed in **S1 File**. One of our unchallenged hemocyte samples showed elevated immune gene expression. To account for a possible unrelated infection, we reduced unwanted variation from this sample as described in the methods.

**Fig 1. Experimental design of RNA sequencing experiment.** Total RNA was extracted from whole larvae or hemocytes recovered from *Hml*$^{\Delta}$.*ds-red.nls* fluorescent larvae. Larvae were left unchallenged or challenged by a systemic infection with *Escherichia coli* or *Staphylococcus aureus* and incubated at 29˚C for 45 minutes. Hemocytes were collected in PBS, on ice, and were immediately sorted FACS and processed for RNA extraction.

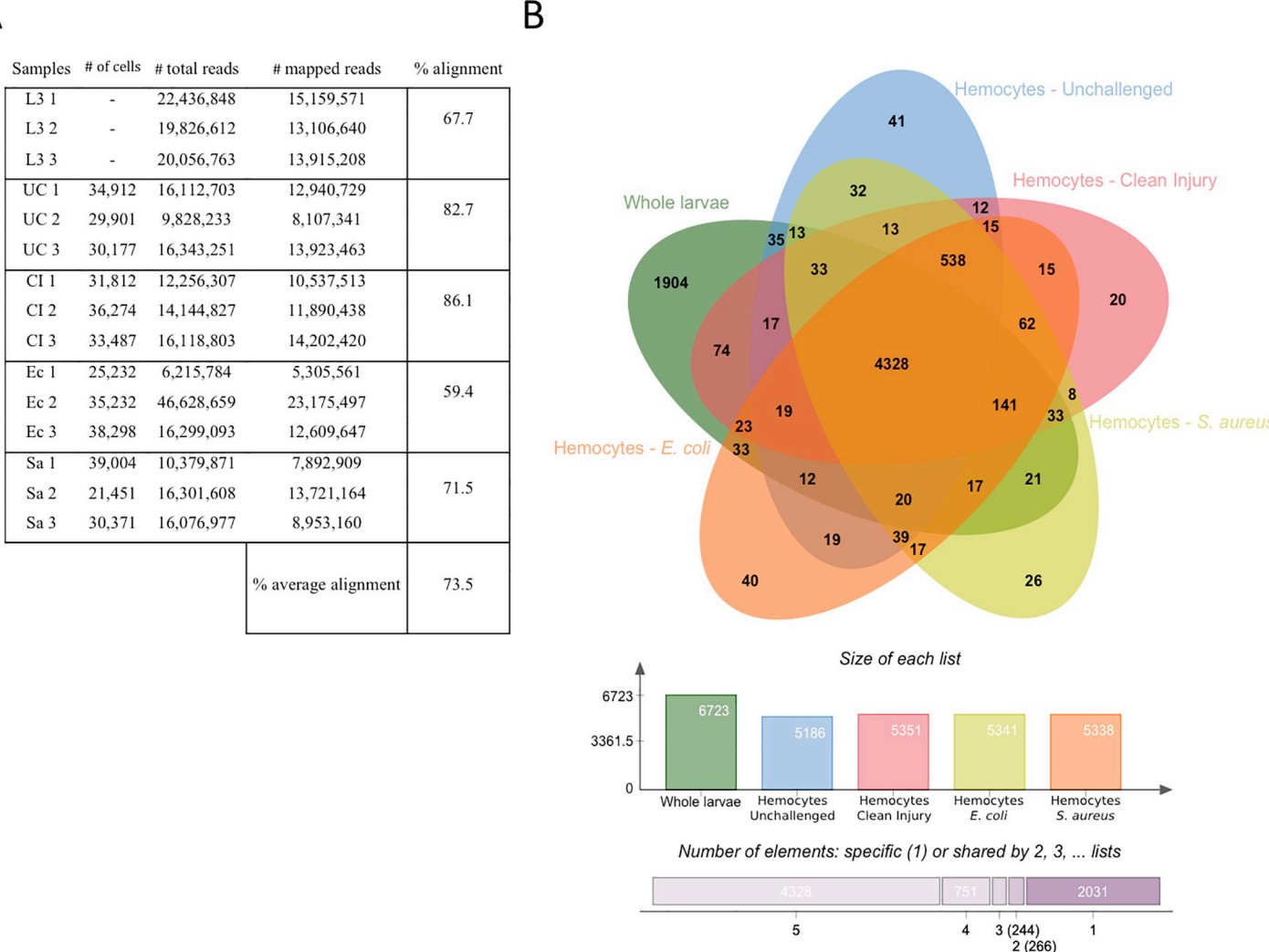

**Fig 2. Transcriptome summaries from unchallenged whole larvae and hemocytes from unchallenged and infected larvae.** (A) Transcriptome summary showing the number of reads for each triplicate in all experimental conditions with their corresponding number of mapped reads and the average percentage of alignment to the *D. melanogaster* genome. (B) Venn diagram representing the quantity of shared genes between all experimental treatments: Unchallenged wandering L3 larvae, hemocytes from unchallenged larvae, hemocytes from clean-pricked larvae (CI), hemocytes from larvae pricked with *Escherichia coli* (Ec), hemocytes from larvae pricked with *Staphylococcus aureus* (Sa).

We then identified differentially expressed genes (DEGs) between all five samples by four pairwise comparisons: "unchallenged hemocytes" samples versus "unchallenged whole larva" samples, "unchallenged hemocytes" samples vs hemocytes from clean injured larva (CIH), CIH vs hemocytes from "*Escherichia coli* infected larvae" (EcH) and CIH vs hemocytes from "*Staphylococcus aureus* infected larvae" (SaH).

## Identification of genes enriched in *Drosophila* larval plasmatocytes

Using our threshold that includes genes with at least 5 reads per 1 million reads, we found that 6,723 genes are expressed in L3 larvae, while unchallenged hemocytes express 5,186 genes. The number of genes expressed in hemocytes is roughly the same as observed by Irving et al. using Affymetrix arrays that identified around 5000 expressed genes in hemocytes [2]. To identify transcripts expressed in the unchallenged hemocyte population, we classified genes according

to their total number of reads (**S1 File**) and their degree of enrichment in Hml+ plasmatocytes compared to whole larvae (**S2 File**, for a selection see **Table 1**). We found that whole larvae and hemocytes shared expression of 4,477 genes, 2,246 genes were uniquely expressed in L3 larvae and 709 genes were uniquely expressed in hemocytes (**Fig 2B**). We confirmed the identity of the plasmatocyte population by the presence of reads for genes known to be specific for plasmatocytes (see below). We found 239 genes encoding transmembrane proteins in unchallenged hemocytes. Of those, 44 were enriched and 195 poorly expressed in hemocytes when compared to whole larvae (**S3 File**). This large set of transmembrane proteins likely contributes to the versatile functions of plasmatocytes. By secreting immune factors, the fat body plays a major role in the humoral response. Plasmatocytes are thought to play a similar role upon infection [42,43]. We therefore looked for genes encoding proteins with a secretion signal in plasmatocytes. We identified 329 such genes expressed in plasmatocytes. Among those, 70 were enriched and 259 were poorly expressed in plasmatocytes when compared to whole larvae (**S4 File** and selection in **Table 1**).

To better characterize gene repertoire of plasmatocytes, we will restrict our analysis to the 5393 genes that were differentially expressed in the whole larvae compared to plasmatocytes (**S2 File**). GO terms analysis (**S1 Fig**) identified many biological processes without clearly highlighting important classes of genes. Thus, we decided to analyze in depth the DEGs identified in our initial analysis.

We first focused our attention on genes known to play a role in *Drosophila* hematopoiesis. We found that genes encoding the transcription factors Serpent [44], U-shaped [45] and Yantar [46], which play a role in pro-hemocyte differentiation, were enriched in plasmatocytes with respective fold changes (FCs) of 7.0, 6.6 and 2.7 compared to whole larvae. We did not identify *glial cells missing* (*gcm*) in our screen, which is consistent with the fact that this gene encodes a transcription factor promoting plasmatocytes maturation only at the embryonic stage [36,47]. The three genes *dome*, *hedgehog* and *Antennapedia*, which positively regulate hematopoiesis in the lymph gland [48], were reduced in circulating plasmatocytes compared to whole larvae, with FCs of -5.3, -173.9 and -2903.3, respectively. Similarly, genes that promote pro-hemocyte maturation in the lymph gland, such as *jumu*, *pyramus*, *thisbe* and *heartless* were also down-regulated in hemocyte samples (with FCs of -2.2, -31.9, -1151.8 and -1655.5, respectively) [49,50]. The gene encoding the transcription factor *collier* (*knot*) that contributes to the lymph gland posterior signaling center [51] was not enriched in plasmatocytes. In contrast, the two genes encoding the transcription factors Pointed and Pannier, which promote hemocyte terminal differentiation [50,52] were enriched in circulating hemocytes with FCs of 3.5 and 15.6 respectively. Finally, genes implicated in crystal cell differentiation such as *Delta*, *serrate* and *notch* [48] were downregulated in plasmatocytes samples compared to the whole larvae samples, with respective FCs of -919.3, -9.9 and -4.5. In contrast, the expression of *lozenge* gene, which encodes the master regulator of crystal cell differentiation, was not affected. The expression of *lozenge* in Hml+ plasmatocytes possibly reflects the trans-differentiation of a subset of them into crystal cells [28]. These results confirmed that collected circulating peripheral plasmatocytes were mostly in the differentiated state. Consistent with this, the *Drosophila* hemocyte marker genes *hemese*, *peroxidasin* and *hemolectin* had respectively 49.0, 15.5 and 6.7 fold higher expression in hemocytes compared to whole larvae samples.

As expected, plasmatocytes were strongly enriched in genes involved in phagocytosis. We found the *scavenger receptor class C, type I* (*ScrCI*), and the Nimrod receptors *Nimrod C1* [53] and *eater* [54] had respective FCs of 48.9, 26.5 and 15.3. We did not identify the *Integrin βv subunit* to be differentially expressed in our screen. Two other Nimrod receptors, *draper* and *simu* (Six-Microns-Under, also named Nimrod C4) can bind phosphatidylserine on dying

**Table 1. Selected DEGs of interest with Fold-changes of unchallenged hemocytes vs. whole larvae from S2 File.**

| CG | Full name | FC | CG | Full name | FC |
|---|---|---|---|---|---|
| **Hemocyte marker** | | | **Stress response** | | |
| CG31770 | Hemese | 49.05 | CG31359 | Heat-shock protein-70Bb | 55.07 |
| CG3978 | pannier | 15.63 | CG4183 | Heat-shock protein-26 | 7.69 |
| CG12002 | Peroxidasin | 15.51 | CG11637 | Ninjurin B | 5.96 |
| CG16707 | visgun | 13.62 | CG4463 | Heat-shock protein-23 | 4.43 |
| CG3992 | serpent | 7.06 | CG12101 | Heat-shock protein-60A | 4.36 |
| CG7002 | Hemolectin | 6.76 | CG4466 | Heat-shock protein-27 | 4.27 |
| CG2762 | u-shaped | 6.66 | CG5436 | Heat-shock protein-68 | 3.48 |
| CG18426 | yantar | 2.73 | CG4147 | Heat shock 70-kDa protein cognate 3 | 3.3 |
| **Immune response** | | | CG34246 | Heat shock protein cognate 20 | 2.68 |
| **Antimicrobial defense** | | | CG1242 | Heat shock protein 83 | 2.59 |
| CG4437 | PGRP-LF | 3.49 | **Oxidative stress response** | | |
| CG4432 | PGRP-LC | 3.23 | **Glutathione S transferase family** | | |
| CG5576 | Immune deficiency | 3.29 | CG6776 | Glutathione S transferase O3 | 10.43 |
| CG8995 | PGRP-LE | 2.78 | CG10045 | Glutathione S transferase D1 | 8.55 |
| CG11992 | Relish | 2.59 | CG17523 | Glutathione S transferase E2 | 8.51 |
| CG32042 | PGRP-LA | 2.27 | CG12242 | Glutathione S transferase D5 | 5.22 |
| CG6134 | spatzle | 10.5 | CG4371 | Glutathione S transferase D7 | 4.89 |
| CG5974 | pelle | 5.31 | CG4381 | Glutathione S transferase D3 | 3.32 |
| CG10520 | tube | 3.13 | CG17530 | Glutathione S transferase E6 | 3.12 |
| CG7629 | Attacin-D | 7.8 | CG11784 | Glutathione S transferase E13 | 3.09 |
| **Melanization** | | | CG5224 | Glutathione S transferase E11 | 2.79 |
| CG42640 | Prophenoloxidase 3 | 25.59 | CG10091 | Glutathione S transferase D9 | 2.46 |
| CG18550 | yellow-f | 13.55 | CG5164 | Glutathione S transferase E1 | 2.35 |
| CG1102 | Melanization protease 1 | 11.28 | CG9362 | Glutathione S transferase Z1 | 2.1 |
| CG8193 | Prophenoloxidase 2 | 9.1 | CG30000 | Glutathione S transferase T1 | 2 |
| CG11331 | Serpin 27A | 7.93 | **Others genes** | | |
| CG7219 | Serpin 28Dc | 3.27 | CG32495 | Glutathione synthetase 2 | 20.18 |
| **Antiviral immunity** | | | CG17753 | Copper chaperone for superoxide dismutase | 3.62 |
| CG7138 | r2d2 | 2.22 | CG8905 | Superoxide dismutase 2 (Mn) | 3.57 |
| **Phagocytic receptors and markers** | | | CG11793 | Superoxide dismutase 1 | 2.8 |
| CG4099 | Scavenger receptor | 48.99 | CG1274 | thioredoxin peroxidase 2 | 2.47 |
| CG8942 | Nimrod C1 | 26.55 | CG1633 | thioredoxin peroxidase 1 | 2.03 |
| CG6124 | eater | 15.35 | **Metabolism** | | |
| CG2086 | draper | 3.19 | **Lipid metabolism / Peroxisome** | | |
| CG10106 | Tetraspanin 42Ee | 3.48 | CG7291 | Niemann-Pick type C-2a | 7.59 |
| CG4591 | Tetraspanin 86D | 4 | CG3083 | Peroxin 19 | 6.67 |
| CG6120 | Tetraspanin 96F | 2.9 | CG3639 | Peroxin 12 | 6.06 |
| CG10742 | Tetraspanin 3A | 3.2 | CG7081 | Peroxin 2 | 4.44 |
| **Opsonins** | | | CG7864 | Peroxin 10 | 2.94 |
| CG33119 | Nimrod B1 | 23.27 | CG3947 | Peroxin 16 | 2.41 |
| CG33115 | Nimrod B4 | 17.69 | CG4289 | Peroxin 14 | 2.2 |
| CG16873 | Nimrod B5 | 3.78 | CG3415 | peroxisomal Multifunctional enzyme type 2 | 2.27 |
| CG2958 | lectin-24Db | 33.93 | CG6783 | fatty acid binding protein | 9.73 |
| CG7106 | lectin-28C | 24.47 | CG4280 | croquemort | 4.74 |
| CG18096 | Thioester-containing protein 1 | 19.2 | **Adenosine metabolism** | | |
| CG5210 | Imaginal disc growth factor 6 | 6.6 | CG5992 | Adenosine deaminase-related growth factor A | 6.59 |

*(Continued)*

**Table 1.** (Continued)

| | | | | | |
|---|---|---|---|---|---|
| CG4472 | *Imaginal disc growth factor 1* | 5.91 | CG11994 | *Adenosine deaminase* | 5.59 |
| Cellular encapsulation | | | Glucose metabolism | | |
| CG14225 | *eye transformer* | 17.16 | CG14816 | *Phosphoglycerate mutase 5* | 4.87 |
| CG3715 | *SHC-adaptor protein* | 7.03 | CG6453 | *Glucosidase 2 beta subunit* | 2.11 |
| CG7830 | *Oligosaccharide transferase gamma subunit* | 2.68 | CG7010 | *Pyruvate dehydrogenase E1 alpha subunit* | 2.06 |
| | | | CG30410 | *Ribose-5-phosphate isomerase* | 2.04 |

cells and promote apoptotic cell internalization, a process called efferocytosis [55,56]. In our screen, we found *draper* transcripts enriched in plasmatocytes samples with a FC of 3.1 whereas *simu* expression was unchanged. Hml+ plasmatocytes were also enriched in genes encoding opsonins such as *Tep1* (FC: 19.2) which has been shown to promote bacterial internalization [57,58]. In contrast, *Tep2* and *Tep3* were down-regulated in hemocytes compared to the whole larvae samples (FCs: -8.4 and -63.1). Furthermore, several genes encoding secreted components of the Nimrod family were enriched in plasmatocytes, most notably *Nimrod B1* and *Nimrod B4* and to a lesser extent *Nimrod B5* (FCs: 23.2, 17.6 and 3.7) as well as *hemese*. Genes encoding secreted Nimrod genes and *hemese* are clustered in the genome together with genes coding for phagocytic receptors such as *Nimrod C1*. Recently, NimB5 has been shown to regulate plasmatocyte adhesion and proliferation [15]. The two plasmatocyte-enriched secreted Nimrod proteins, NimB1 and NimB4, are promising candidate genes regulating important plasmatocyte functions, possibly phagocytosis [59]. *Drosophila* plasmatocytes were also enriched in several cytoskeleton proteins (see in **S1 Table**) such as SCAR, which has been shown contribute to phagocytosis and cell migration [60]. The two GTPases, Rac1 and Rac2, which have been implicated in phagocytosis and cellular response were also enriched in plasmatocytes (FCs: 2.8 and 4.1) [61–63]. A recent study revealed an important role of peroxisomes in phagocytosis and immunity [64]. Consistent with this, several peroxins that encode components of peroxisomes (peroxins 19, 12, 2, 11, 10, 16, and 14) were strongly enriched in plasmatocytes. In addition, several genes encoding Tetraspanins were enriched in plasmatocytes. Tetraspanins are implicated in a wide range of functions in *Drosophila* such as protein stabilization at the plasma membrane and cell signaling regulation, and could contribute to phagocytosis or adhesion [65]. Specifically, we identified *late bloomer* and the *Tetraspanins 86D*, *42Ee*, *3A* and *96F* with respective FCs ranging from 2.8 to 3.9. Collectively, the enrichment of genes encoding phagocytic receptors, opsonins, and cytoskeletal proteins in plasmatocytes confirm their phagocytic ability.

The systemic antimicrobial response, which encompasses the production and release of many immune effectors into the hemolymph, is regulated by two NF-κB pathways, namely Imd and Toll [66]. There is strong evidence that these two pathways are functional in hemocytes [67]. We found that plasmatocytes are enriched in several components of the Imd pathway, notably Imd and Relish. The genes encoding three transmembrane receptors of the Peptidoglycan recognition proteins (PGRP) family, PGRP-LF, PGRP-LC, and PGRP-LA, which are organized in a cluster in the *Drosophila* genome and contribute to Imd pathway activity, were also higher in plasmatocytes (FCs: 3.4, 3.2 and 2.2 respectively) [68,69]. The gene encoding the intracellular pattern recognition PGRP-LE that is involved in the sensing of monomeric peptidoglycan of Gram-negative bacteria and autophagy was also increased in plasmatocytes (FC: 2.7) [70–72]. We also confirmed that plasmatocytes have an increased expression of the gene *spatzle* (FC: 10.5), which encodes the ligand of the Toll pathway [66,73] as well as the genes encoding the adaptor Tube and the kinase Pelle (respective FCs: 3.1 and 5.3).

Crystal cells are the main hemocyte type involved in the melanization reaction, expressing both Prophenoloxidases 1 (PPO1) and 2 (PPO2) [9]. In contrast, Prophenoloxidase 3 (PPO3) is expressed in lamellocytes [10]. Surprisingly, we found that both *PPO2 and PPO3* were enriched in circulating cells with fold changes of 9.1 and 25.5 compared to whole larvae. Their expression in plasmatocytes could reflect the transdifferentiation of Hml positive cells into crystal cells and lamellocytes or an unexpected contribution of plasmatocytes in the production of prophenoloxidase. Plasmatocytes had increased expression of genes encoding for several of the enzymes that have been linked to prophenoloxidase activity (e.g. *yellow f*, FC: 13.55; see **Table 1**). Complex serine protease cascades regulate important immune functions (i.e. melanization, Toll) in the hemolymph. We found that plasmatocytes have higher expression of the gene encoding the serine protease MP1 (FC: 11.2), which regulates melanization, and serpins such as *Serpin27A* (FC: 7.9) and *Serpin28Dc* (FC: 3.2), which negatively regulate melanization and Toll [74–77]. The immune function of lectins is poorly characterized in *Drosophila* but some of them have been implicated in immunity in other insects [2,78,79]. We found that two secreted lectins, lectin-24Db and lectin-28C were strongly enriched in plasmatocytes (FCs: 33.9 and 24.4).

Plasmatocytes also showed increased expression of many genes involved in the oxidative stress response, notably many glutathione S transferase genes and other detoxifying enzymes such as peroxidasin (FC 15.51), superoxide dismutases 1 and 2 (FCs: 2.8 and 3.5) or thioredoxin peroxidase 1 and 2 (FC: 2 and 2.4). These genes may play a role in immune response activation [80]. In parallel, we observed an increase in gene transcripts involved in other cellular stress pathways, e.g. many heat shock proteins and Ninjurin B (FC: 5.9). We cannot exclude the possibility that this expression profile related to cellular stress also reflects a quick response of plasmatocytes to stresses imposed by procedures (i.e. temperature switch from 25˚C to 29˚C and FACS processing), despite maintaining samples on ice following extraction.

Plasmatocytes contribute to the production of the basal membrane in the embryo [81]. In agreement with this, larval plasmatocytes had increased expression of genes encoding components of the basal membrane, notably Laminins A, B1 and B2 (FCs: 5.7, 7.2 and 8.7) [8], Viking (collagen IV), and secreted enzymes that contribute to basal membrane formation (fat-spondin, Glutactin, Peroxidasin, Matrix metalloproteinase 2 with FCs of 19.1, 18.6, 15.5 and 4.9). Consistent with previous studies that have suggested an important role of plasmatocytes in adenosine metabolism at the larval stage [82] we found that the two genes *Adenosine deaminase-related growth factor A* (*ADGF-A*) and *Adenosine deaminase* were upregulated (FCs: 6.59 and 5.59) while *Adenosine deaminase-related growth factor A2* and *D* were downregulated. Recent studies have also highlighted a significant contribution of plasmatocytes in lipid uptake and storage, complementary to the fat body [83]. Consistent with this, the gene *croquemort*, which encodes a lipid binding receptor of the CD36 family, is also enriched in plasmatocytes (FC: 4.7) [84]. The *fatty acid binding protein* (*fabp*) gene was enriched 9-fold in plasmatocytes. Finally, our transcriptome analysis shows that plasmatocytes express many transporters, the aquaporin Prip being one of the most enriched compared to other tissues (FC: 12.4).

## The early plasmatocyte response to clean injury

We then explored the transcriptomic response of plasmatocytes to clean injury in further detail by comparing the transcriptome of hemocytes extracted from unchallenged larvae to hemocytes from larvae 45 minutes after clean injury. We choose this time point as it corresponds to the time necessary to fully internalize or phagocytose a particle [85]. We identified 664 DEGs after clean injury compared to hemocytes from unchallenged larvae. Among these genes, 358 were up-regulated and 306 were down-regulated in the clean injury samples using a two-fold

change criteria and p<0.05 cut-off (**S5 File,** See selection in **Table 2**). We then proceeded to GO terms analysis by focusing on genes that are upregulated in hemocytes after clean injury compared to unchallenged hemocytes, and that have a P-value < 0.05. We identified several GO-term groups significantly enriched upon clean injury. Genes with GO terms for molecular function assigned to cell metabolism, actin mobilization and cytoskeleton organization, anti-oxidant and stress responses were particularly affected (**S2 Fig**).

We found a significant increase in transcripts corresponding to *Glutathione-S-transferase* (GST) genes (D2, E1, D3, E3, E8, D7, D5, E6) which encode antioxidants enzymes that detoxify hydrogen peroxide and lipid peroxides [86,87]. Genes encoding three Cytochrome P450 enzymes (Cyp6a20, Cyp6a17 and Cyp6a23) and the ABC transporter Multidrug resistance protein 4 (Mrp4), which are involved in detoxification, were also enriched upon clean injury (FC: 10.5, 3.1, 2.8 and 2.8).

Other stress responsive genes such as *Frost*, *Hsp70*, *Hsp68*, *Ninjurin A*, the Hsp co-factor *starvin* and *DnaJ-like-1* were also more highly expressed in clean injury conditions. Consistent with this stress response, the JNK stress responsive pathway was activated as evidenced by an increase expression of *puckered* (*puc*, FC: 2.3).

Many other upregulated genes such as *Gadd45*, *kayak*, *Larval cuticle proteins 1*, *3* and *4*, and *Matrix metalloproteinase 1* which play a key role in wound healing, extracellular matrix generation and cuticle repairing, were also more highly expressed in plasmatocytes [88,89].

Our study confirms that the JAK-SAT ligand gene *upd3* which orchestrates the systemic wound response [3,90] was upregulated in plasmatocytes upon clean injury (FC: 4.2). The gene encoding Wallenda, a MAP3K that regulates stress response by regulating the expression of the Materazzi lipid binding protein gene in Malpighian tubules [91], was also upregulated (FC: 2.6).

It is well established that clean injury, in the non-sterile condition used in this study, triggers a transient and weak antimicrobial response [92]. We found that genes encoding components of the immune responsive pathways Imd (*Relish*, *PGRP-LB*, *PGRP-LF*) and the Toll pathways (*cactus)* were upregulated (respective FCs: 4.5, 3.2, 2.4 and 2.1). A subset of antimicrobial peptide genes, notably *Cecropin B*, *A2* and *C* (respective FCs: 13, 8.3 and 6.8) and *Attacin-B* (FC: 6.8), were upregulated by clean injury.

Interestingly, two genes whose mutations have been associated with refractoriness to virus C and Sigma, *pastrel* and *ref(2)P* respectively, were upregulated upon clean injury (FCs: 3.1 and 2.0) [93]. While the function of *ref(2)P* in autophagy is well established [94], the role of *pastrel* is poorly characterized. The induction of *pastrel* in plasmatocytes suggests that it could play an important function in activated plasmatocytes, as the result of a potential viral infection.

Genes such as *fondue* (FC: 4.2), *Larval-serum protein 1γ* (FC: 8.3) and *Hemolectin* (FC: 2.2) that are implicated in hemolymph clotting [95] were up-regulated in clean-injured samples. Of note, *dopa decarboxylase* and the GTP cyclohydrolase *punch*, genes which encode enzymes that regulate melanin formation, were enriched in clean injury samples confirming the contribution of plasmatocytes to the melanization process [96]. Collectively, our study shows that plasmatocytes contribute to wound healing by inducing genes involved in stress response, ROS detoxification and cytoskeletal remodeling. The induction of genes encoding components of the Toll and Imd immune signaling pathways may reinforce the reactivity of these immune cells.

Another major class of genes upregulated upon clean injury are those involved in the cytoskeleton (**S1 Table**). This includes genes involved in actin remodeling, microtubule formation and adhesion that likely reflect the change of shape observed in 'activated plasmatocytes' that are known to be more adhesive and display filopodia [97]. Among them, were the *Integrin*

 *Drosophila* blood cells comparative RNA-Seq analyses in response to clean injury and septic injury

**Table 2. Selected DEGs of interest with Fold-changes of clean injury hemocytes vs. unchallenged hemocytes from S5 File.**

| CG | Full name | FC | CG | Full name | FC |
|---|---|---|---|---|---|
| Immune response | | | Gene regulation | | |
| Antimicrobial defense | | | CG13194 | *pyramus* | 31.23 |
| CG11992 | *Relish* | 4.51 | CG10045 | *Daughters against dpp* | 4.82 |
| CG43720 | *sickie* | 4.12 | CG33542 | *unpaired 3* | 4.26 |
| CG14704 | *Peptidoglycan recognition protein LB* | 3.21 | CF13780 | *PDGF- and VEGF-related factor 2* | 3.87 |
| CG4437 | *Peptidoglycan recognition protein LF* | 2.44 | CG4371 | *wallenda (MAP Kinase Kinase)* | 2.60 |
| CG1399 | *Leucine-rich repeat* | 2.62 | CG32406 | *PVR adaptor protein* | 2.53 |
| CG16712 | *Immune induced molecule 33* | 2.54 | CG15154 | *Suppressor of cytokine signaling at 36E* | 2.52 |
| CG5848 | *cactus* | 2.18 | CG7850 | *puckered* | 2.30 |
| CG1878 | *Cecropin B* | 13.04 | Cytoskeleton organization | | |
| CG1367 | *Cecropin A2* | 8.38 | CG3259 | *Intraflagellar transport 54* | 14.41 |
| CG1373 | *Cecropin C* | 6.85 | CG11798 | *charlatan* | 7.49 |
| CG18372 | *Attacin-B* | 6.86 | CG4843 | *Tropomyosin 2* | 7.46 |
| Clotting | | | CG5372 | *Integrin alphaPS5 subunit* | 7.19 |
| CG15825 | *fondue* | 4.23 | CG12008 | *karst* | 6.75 |
| CG7002 | *Hemolectin* | 2.23 | CG5695 | *jaguar* | 6.04 |
| Phagocytosis | | | CG43976 | *Rho guanine nucleotide exchange factor 3* | 4.98 |
| CG31962 | *Scavenger receptor* | 2.74 | CG1212 | *p130CAS* | 4.85 |
| Melanization | | | CG8865 | *Ral guanine nucleotide dissociation stimulator-like* | 4.15 |
| CG10697 | *Dopa decarboxylase* | 15.09 | CG10076 | *spire* | 4.08 |
| CG9441 | *Punch* | 3.32 | CG33103 | *Papillin* | 4.06 |
| Repair | | | CG13503 | *Verprolin 1* | 3.60 |
| CG2043 | *Larval cuticle protein 3* | 19.21 | CG3937 | *cheerio* | 3.28 |
| CG4859 | *Matrix metalloproteinase 1* | 9.77 | CG11949 | *coracle* | 3.24 |
| CG2044 | *Larval cuticle protein 4* | 3.84 | CG14396 | *Ret oncogene* | 3.18 |
| CG11086 | *Growth arrest and DNA damage-inducible 45* | 2.57 | CG33558 | *missing-in-metastasis* | 3.06 |
| CG33956 | *kayak* | 2.18 | CG2184 | *Myosin light chain 2* | 3.02 |
| Others | | | CG10119 | *Lamin C* | 2.73 |
| CG8588 | *pastrel* | 3.10 | CG18214 | *trio* | 2.68 |
| CG6821 | *Larval serum protein 1 gamma* | 8.35 | CG33694 | *CENP-ana* | 2.58 |
| Stress response | | | CG5164 | *cappuccino* | 2.41 |
| Heat shock proteins | | | CG9362 | *kugelei* | 2.40 |
| CG5834 | *Heat-shock protein-70Bbb* | 16.44 | CG10522 | *sticky* | 2.38 |
| CG6489 | *Heat-shock protein-70Bc* | 15.67 | CG18076 | *short stop* | 2.34 |
| CG18743 | *Heat-shock-protein-70Ab* | 10.25 | CG1560 | *myospheroid* | 2.12 |
| CG31359 | *Heat-shock protein-70Bb* | 8.78 | CG42274 | *Rho GTPase activating protein at 18B* | 2.11 |
| CG5436 | *Heat-shock protein-68* | 6.28 | CG1520 | *WASp* | 2.1 |
| Cyp450 | | | Vesicle trafficking | | |
| CG10245 | *Cyp6a20* | 10.59 | CG8024 | *Rab32* | 3.15 |
| CG10241 | *Cytochrome P450-6a17* | 3.18 | CG14001 | *blue cheese* | 2.34 |
| CG10242 | *Cyp6a23* | 2.83 | CG14296 | *Endophilin A* | 2.15 |
| Others | | | Cell death effectors | | |
| CG9434 | *Frost* | 9.19 | CG1600 | *Death resistor Adh domain containing target* | 3.63 |
| CG32130 | *starvin* | 8.80 | CG33134 | *Death executioner Bcl-2* | 2.33 |
| CG12703 | *Peroxisomal Membrane Protein 70 kDa* | 4.61 | Metabolism | | |
| CG14709 | *Multidrug resistance protein 4* | 2.85 | Glucose metabolism | | |
| CG10578 | *DnaJ-like-1* | 2.54 | CG5932 | *magro* | 26.43 |

*(Continued)*

 

**Table 2.** (Continued)

| CG | Full name | FC | CG | Full name | FC |
|----|-----------|----|----|-----------|----|
| *CG6449* | *Ninjurin A* | 2.46 | *CG34360* | *Glucose transporter 4 enhancer factor* | 8.97 |
| *CG31216* | *Nicotinamide amidase* | 2.44 | *CG5889* | *Malic enzyme b* | 4.28 |
| Oxidative stress response | | | *CG4625* | *Dihydroxyacetone phosphate acyltransferase* | 4.24 |
| Glutathione S transferase family | | | *CG6906* | *Carbonic anhydrase* | 3.70 |
| *CG4181* | *Glutathione S transferase D2* | 10.36 | *CG8256* | *Glycerophosphate oxidase 1* | 2.69 |
| *CG5164* | *Glutathione S transferase E1* | 5.60 | Lipid catabolism | | |
| *CG4381* | *Glutathione S transferase D3* | 4.19 | *CG3620* | *no receptor potential A* | 2.25 |
| *CG17524* | *Glutathione S transferase E3* | 4.19 | *CG11055* | *Hormone-sensitive lipase* | 2.22 |
| *CG17533* | *Glutathione S transferase E8* | 3.21 | *CG1882* | *pummelig* | 2.05 |
| *CG4371* | *Glutathione S transferase D7* | 3.20 | Amino acid storage | | |
| Others | | | *CG6821* | *Larval serum protein 1 gamma* | 8.35 |
| *CG2259* | *Glutamate-cysteine ligase catalytic subunit* | 3.56 | *CG4178* | *Larval serum protein 1 beta* | 3.54 |
| | | | *CG2559* | *Larval serum protein 1 alpha* | 2.69 |

*alphaPS5 subunit* gene (FC: 7.1) [98] and the integrins *charlatan* and *myospheroid* (FCs: 7.4 and 2.1) [99,100] which are induced upon lamellocyte differentiation. Interestingly, *Integrin βv subunit*, a gene that encodes a transmembrane protein implicated in apoptotic corpses clearance in embryonic hemocytes [101], and *Scavenger receptor class C, type III* (*Sr-CIII*) were also induced (FCs: 3.9 and 2.7). Moreover, the gene encoding the FGF ligand Pyramus that has been shown to promote blood cell progenitors differentiation in the lymph gland [50] was the most up-regulated gene 45 minutes after clean injury. This suggests that FGF-R pathway activation in plasmatocytes by the ligand Pyramus could play a prominent role in promoting the differentiation of peripheral plasmatocytes upon injury, akin to the process observed in the lymph gland. On this line, Tattikota *et al.* observed an enrichment of transcripts encoding the FGF ligand Branchless and its receptor Breathless in crystal cells and lamellocytes subsets respectively, and revealed a role of the FGF pathway in the encapsulation of parasitoid wasps [37]. This also highlights the importance of the FGF pathway in cell specification. Two genes from the PVR (PDGF/VEGF-related factor) pathway that are implicated in hemocyte survival and migration [102–104], one encoding the PVR adaptor PVRAP and the other the PVR ligand Pvf2, were up-regulated upon clean injury (FCs: 2.5 and 3.8). Consistent with this, we also observed that clean injury triggers the down-regulation of apoptosis-associated genes such as *head involution defective* (*hid*, FC: 15.7) [105] and *Deneddylase 1* (*Den1*, FC: 2.2) [106]. This suggests that wounding stimulates blood cell survival as well as blood cell pool expansion.

In mammals, macrophages undergo massive metabolic change upon activation [107,108]. Notably, lipid catabolism and glucose consumption are essential components of mammalian macrophage activation in order to fuel the cell as well as to produce inflammatory mediators [109]. We next investigated whether clean injury also induces a metabolic reprogramming in plasmatocytes. Interestingly, when analyzing the GO terms enrichment, we found an overrepresentation of "lipase activity" related genes (**S2 Fig**). Indeed, we observed the up-regulation of the *magro*, *alpha/beta hydrolase2* (*Hydr2*), the *phospholipase c at 21c* and *no receptor potential A* (*norpA*) genes with respective FCs of 26.4, 2.9, 2.6 and 2.2. Upregulation of *apolipophorin* (*apoLpp*) and *ATP binding cassette subfamily A* (*ABCA*) genes (FCs: 2 and 4.2) indicate that both lipid catabolism and lipid uptake are induced upon clean injury in plasmatocytes, which may fuel the increased energy demand of the activated cells. In agreement with the concept of metabolic reprogramming, we noted the up-regulation of the *Glucose transporter 4 enhancer factor* gene (*Glut4EF*, FC 8.9) [110], a transcription factor regulating the *Glucose*

*transporter 4* gene [111], also known as *solute carrier family 2 member 4*), and of the *Glycogen phosphorylase* (GlyP) gene coding for the enzyme catalyzing the rate-limiting step of glycogenolysis. The induction of these two genes suggests that upon injury plasmatocytes may increase their metabolic activity by increasing glucose provisioning. Genes of the mTOR signaling pathway, that is known to stimulate a glycolytic metabolism, were also upregulated: *thor*, *rictor* and *phosphoinositide-dependent kinase 1* (*pdk1*) and (FCs: 2.7, 2.5 and 2.0). Additionally, the *men* gene encoding malate deshydrogenase, which is known to sustain active glycolysis by replenishing the cytosolic NAD pool and by limiting tricarboxylic acid cycle (TCA) refueling [112], was also upregulated.

## Plasmatocytes gene expression signature in response to bacterial infection

Finally, we explored the transcriptional response of blood cells upon septic injury with *Escherichia coli* (EcH samples, **Table 3** and **S6 File**) and *Staphylococcus aureus* (SaH samples, **Table 4** and **S7 File**) and compared it with the transcriptional profile of hemocytes from clean-injured larvae. We were interested to know if the presence of bacteria affects plasmatocyte response and whether hemocytes react in a different way to infection by Gram-negative versus Gram-positive bacteria (**Fig 3**). Studies have shown that *E. coli* is an efficient inducer of the Imd pathway and is sensitive to the action of antibacterial peptides [113]. In contrast, *S. aureus*, as a lysine-type bacterium, is a potent inducer of the Toll pathway [114] and is combatted by Toll mediated production of Bomanin [113], melanization [77], and phagocytosis [115]. Interestingly, we identified 104 and 92 uniquely expressed genes in hemocytes from larvae infected with *E. coli* (EcH) and *S. aureus* (SaH) respectively compared to hemocytes from clean injured larvae. In EcH samples, we identified 84 up-regulated genes and 151 down-regulated genes

**Table 3. Selected DEGs of interest with Fold-changes of *E. coli* hemocytes vs. Clean injury hemocytes from S6 File.**

| CG | Full name | FC |
|---|---|---|
| Immune related genes—cellular immunity | | |
| CG8175 | *Metchnikowin* | 7.31 |
| CG10794 | *Diptericin B* | 4.97 |
| CG9496 | *Tetraspanin 29Fb* | 4.53 |
| CG1364 | *Cecropin A1* | 4.04 |
| CG1367 | *Cecropin A2* | 3.33 |
| CG32185 | *Elevated during infection* | 3.17 |
| CG10146 | *Attacin-A* | 2.74 |
| CG7629 | *Attacin-D* | 2.72 |
| CG16844 | *Bomanin Short 3* | 2.67 |
| CG1373 | *Cecropin C* | 2.52 |
| CG14704 | *Peptidoglycan recognition protein LB* | 2.07 |
| Cell migration / cytoskeleton reorganization | | |
| CG31004 | *mesh* | 3.47 |
| CG6976 | *Myosin 28B1* | 2.03 |
| Stress response | | |
| CG9434 | *Frost* | 2.99 |
| CG3050 | *Cyp6d5* | 2.42 |
| Glucose metabolism | | |
| CG8693 | *Maltase A4* | 4.59 |
| CG11909 | *Target of brain insulin* | 3.89 |

**Table 4. Selected DEGs of interest with Fold-changes of *S. aureus* hemocytes vs. Clean injury hemocytes from S7 File**.

| CG | Full name | FC |
|---|---|---|
| Immune related genes—cellular immunity | | |
| CG32185 | *elevated during infection* | 3.96 |
| CG8175 | *Metchnikowin* | 3.57 |
| CG12143 | *Tetraspanin42Ej* | 2.37 |
| CG15917 | *Growth-blocking peptide 1* | 2.29 |
| CG10794 | *Diptericin B* | 2.19 |
| CG16844 | *Bomanin Short 3* | 2.08 |
| CG12840 | *Tetraspanin 42El* | 2.02 |
| Cell migration / cytoskeleton reorganization | | |
| CG32082 | *Insulin receptor substrate 53 kDa* | 2.71 |
| Extracellular matrix components | | |
| CG6281 | *Tissue inhibitor of metalloproteases* | 2.67 |
| CG42768 | *Muscle-specific protein 300 kDa* | 2.55 |
| Stress response–cell death | | |
| CG4319 | *reaper* | 2.59 |
| CG10391 | *Cyp310a1* | 2.43 |
| CG8453 | *Cyp6g1* | 2.40 |
| Oxidative stress response | | |
| CG12242 | *Glutathione S transferase D5* | 2.02 |

whereas in SaH samples, we identified 103 up-regulated genes and 81 down-regulated genes (see S6 and S7 Files). The two significantly enriched GO component categories upon infection with *E. coli* or *S. aureus* correspond to secreted components (GO term GO:0005615 and GO:0044421) and an "antibacterial humoral response" (GO:0019731) which is consistent with an increased expression of antimicrobial peptides (AMP) upon infection compared to clean injury (Fig 3, S6 and S7 Files, see selections in **Tables 3** and **4**).

Our RNAseq study reveals a small subset of genes that were induced upon both *E. coli* and *S. aureus*, notably the antimicrobial peptide coding gene *Metchnikowin* (*Mtk*, against Gram-positive and fungi, FC 7.3) and surprisingly, many genes annotated as 18S or 28S ribosomal RNA pseudogenes. Challenge with *E. coli* leads to specific induction of several Imd target genes, notably the antibacterial peptides *Diptericin B*, *Cecropins A1*, *A2 and C*, *Attacins A and D*, as well as *PGRP-LB*, a gene encoding a negative regulator of the Imd pathway that scavenges peptidoglycan [116]. Another immunity gene, *edin*, was also upregulated in EcH samples compared to clean injury samples (FC: 3.1). *Edin* has previously been described as upregulated in S2 cells upon *E. coli* infection and is needed for the increase in plasmatocyte numbers and for the release of sessile hemocytes into the hemolymph upon wasp infection [117]. Thus, the increase in *edin* could reflect the mobilization of sessile hemocytes into circulation. It is important to note the down-regulation of several heat-shock protein genes such as *Hsp27*, *Hsp70ab*, and *Hsp70Bc* in the EcH samples (FCs: -2.0, -3.0 and -3.0). Thus, septic injury with *E. coli* tends to orient the hemocyte towards an antibacterial response while clean injury directs a stress and repair response.

As in EcH samples, SaH samples also show an enrichment of GO processes associated with the immune response, such as the up-regulation of the *Metchnikowin* gene but also *Diptericin B* (FC: 3.57 and 2.19) and one Bomanin gene: *Bomanin Short 3* (*BomS3*, FC: 2.0). The specific induction of antibacterial peptide genes (*AttD*, *Cec*), known to be regulated by the Imd pathway in response to *E. coli* but not *S. aureus* indicates that the hemocytes can mount a

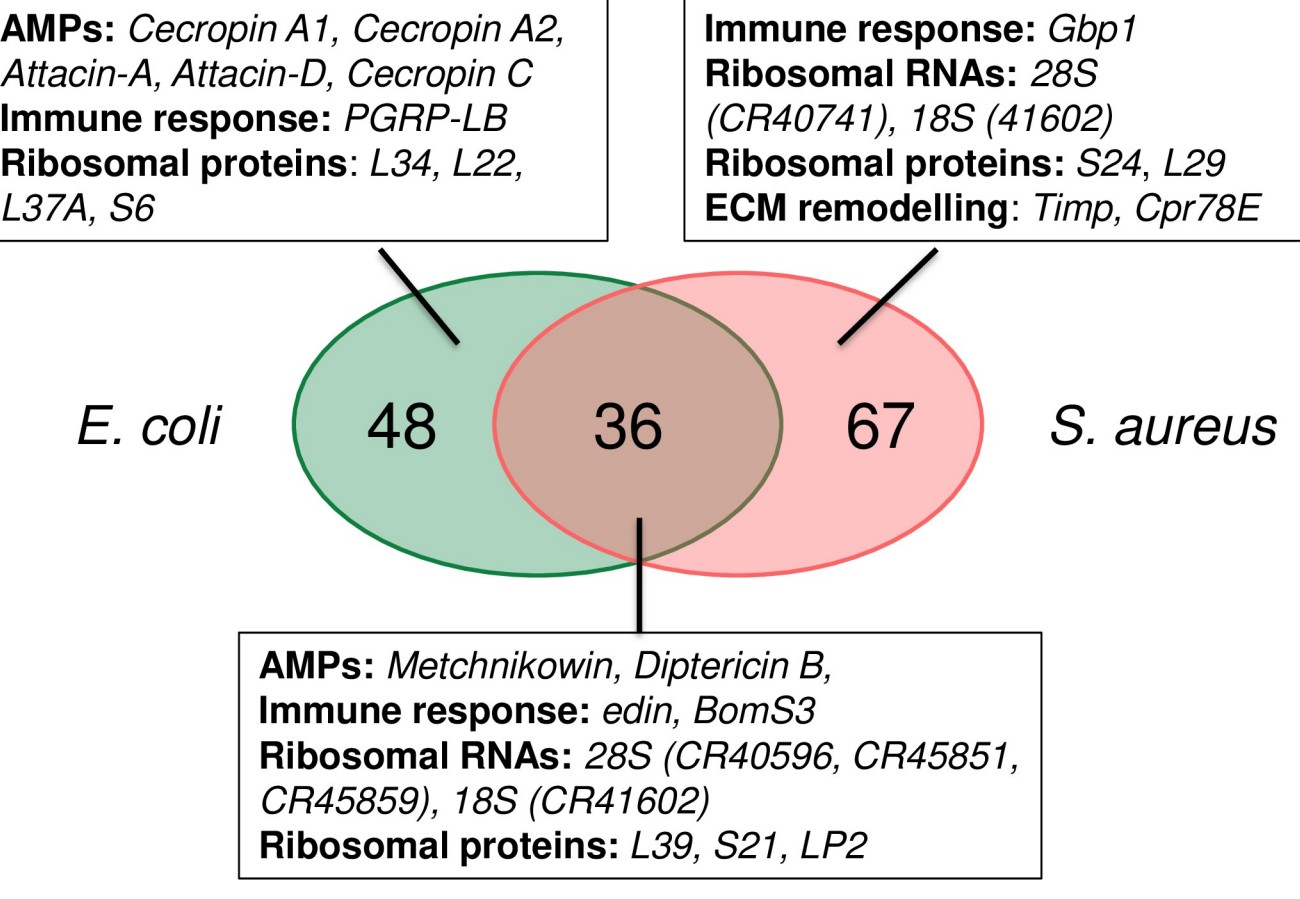

**Fig 3. Genes induced in hemocytes upon bacterial infection.** Venn diagram illustrating genes that were enriched in hemocytes from larvae pricked with *Escherichia coli* (*E. coli*) or *Staphylococcus aureus* (*S. aureus*) compared to clean-pricked larvae (CI).

differentiated response to these two bacteria within 45 minutes. We also observed an increase in expression of *Growth-blocking peptide 1* (*Gbp1*). Gbp1 was characterized as a cytokine that plays a role in IMD activation upon Gram-negative bacterial challenge in both the fat body and hemocytes [118]. Our study rather points to a specific induction of *Gbp1* in response to a Gram-positive bacterial challenge. Several genes encoding proteins that could function in phagocytosis such as *Biogenesis of lysosome-related organelles complex 1, subunit 2,* and *Tetraspanin 42El* were specifically enriched by two-fold in *S. aureus* versus clean injury. Several genes implicated in cell division were down-regulated in *S. aureus* samples compared to clean injury samples, such as *mitotic spindle* and *nuclear protein* (*mink*, -2.4 fold change) [119], *stathmin* (*stai*, -3.8 fold change) [120] and *cyclin B* (*cycB*, -2.2 fold change) [121] suggesting a cell cycle arrest in response to infection with this Gram-positive bacterium (**S7 File**) [122]. Surprisingly, the gene encoding the lipase Magro was expressed 14 times less upon systemic infection with *S. aureus* compared to clean injury or EcH. Thus, at the 45 minutes time point, a challenge by *S. aureus* tends to orient plasmatocytes towards a lower production of secreted factors and decreased lipid catabolism. We hypothesize that plasmatocytes contribute to host defense in different ways against *S. aureus* and *E. coli*. Phagocytosis of *S. aureus* may not be compatible with cell division. Decreased lipase activity may reflect a reduced energy demand of these plasmatocytes compared to energetically expensive AMP production in those infected with *E. coli*.

## Discussion

The *Drosophila* immune response has been the focus of extensive genome-wide gene expression studies that open the route to successful post-genomic functional characterization of novel immune genes [123]. In contrast, transcriptome studies of hemocytes have been rather limited, or have used S2 or mbn-2 hemocyte-derived cell lines that do not reflect an integrated model [2,124,125]. This was mostly due to the difficulties in collecting enough pure material, as hemocytes represent a tiny fraction of *Drosophila* larvae. Recently, FACS sorting of hemocytes, and the use of new hemocyte-specific markers has facilitated the extraction of plasmatocytes. Taking advantage of this, we performed a comprehensive RNAseq analysis of Hml + plasmatocytes in absence of infection and 45 minutes following clean injury or septic injury with *E. coli* or *S. aureus*. We found that FACS purification did not affect the lineage characteristics of the sample allowing us to use these very pure populations to characterize transcriptomic variation in Hml+ positive plasmatocytes. As Hml+ cells are widely used to study hemocyte function such as adhesion, sessility, metabolism and phagocytosis, our dataset is an important contribution to the community. Our RNAseq study also complements a recent single cell analysis of *Drosophila* hemocytes that has revealed the various states of hemocyte differentiation [36,37,38,39]. It is important to note that our chosen experimental design may have affected the transcriptome of unchallenged hemocytes slightly. We incubated all larvae, previously reared at 25˚C, for 45 minutes at 29˚C after treatment to accelerate the transcriptional response, which might result in a small heat shock induction. Despite this limitation, comparisons between conditions revealed interesting patterns of expression that could be useful to further functional studies of hemocytes.

Our study provides the full repertoire of genes expressed in plasmatocytes and their expression levels, notably those encoding transmembrane proteins and secreted factors (**S3** and **S4 Files**). Screening these genes in future studies will allow a better characterization of plasmatocyte functions in phagocytosis, migration, and sessility and a greater understanding of how these motile cells interact with other tissues. Consistent with a previous Affymetrix based study [2] and a recent single cell study [36], we found that plasmatocytes express a large repertoire of phagocytic receptors, opsonins and cytoskeleton proteins, reflecting their important function as phagocytes. Consistent with older studies that demonstrated that hemocytes produce antimicrobial peptides upon challenge [30,42,126], we confirmed that hemocytes express genes encoding several components of the Toll and Imd pathways, as well as some of their downstream target genes. They have the ability to induce a subset of antimicrobial peptides upon challenge. Use of AMP-reporter genes [42,126] and recent single cell analysis [36] has shown that only a fraction of plasmatocytes express antibacterial peptides upon immune challenge, indicating that some sub-populations of plasmatocytes are specialized for this task. One of the most surprising observations is the high expression of Attacin-D in plasmatocytes, as this antibacterial peptide is devoid of a signal peptide. The precise role of this AMP is unknown, and further studies should decipher whether it is secreted by an atypical mechanism or if it functions intracellularly. Our study confirms that plasmatocytes contribute to clotting and melanization, two important hemolymph immune functions [127]. Surprisingly, our RNAseq analysis detected an unexpected expression of PPO2 and PPO3 in Hml+ plasmatocytes, which were previously shown to be specific to crystal cells and lamellocytes. As single cell analysis studies have confirmed that PPO2 and PPO3 are indeed restricted to crystal cells and lamellocytes [36], the high expression of PPO2 and PPO3 in our sample can be explained by the presence of Hml+ plasmatocytes that are undergoing their transdifferentiation into crystal cells and lamellocytes [27,28,99]. The fat body, particularly at the larval stage, is the main organ producing hemolymphatic proteins. Our study confirms that like the fat body,

Hml+ plasmatocytes also express a large repertoire of secreted proteins, notably components of the extracellular matrix and opsonins. It is currently unclear how the synthesis of hemolymphatic protein is allocated between the fat body and plasmatocytes, or why plasmatocytes are involved in the secretion of molecules such as extracellular matrix components or AMPs. An interesting hypothesis is that plasmatocytes, by virtue of their migratory ability, function as local repairers that can locally supply and enrich specific factors. Indeed, a recent study indicated that a distinct pool of plasmatocytes, the "companion plasmatocytes" expressing collagen IV, are tightly associated with the developing ovaries from larval stages and onward [128]. Eliminating these companion plasmatocytes or specifically blocking their collagen IV expression during larval stages causes abnormal ovarian niches with excess stem cells in adults. This suggests that hemocytes could have short-range action consistent with the notion of local repairers. Deciphering specific roles for the hemocytes and fat body in the production of hemolymphatic proteins is an interesting prospect.

Despite extensive research on *Drosophila melanogaster* blood cell adaptation to wounding [33,129] only rare transcriptomic analyses has been performed, and have specifically analyzed their response to clean injury. One main study performed in 2008 tried to provide insight into specific hemocyte response to wounding. Stramer *et al.* compared transcriptional responses of wild-type embryos and *serpent* null embryos, which are devoid of hemocytes, 45 minutes following wounding. They identified a limited number of significantly affected genes, probably because of a dilution effect due to the limited number of hemocytes overall in embryos [88]. However, they identified the up-regulation of secreted phospholipase A2 that plays a role in the production of eicosanoids, key signaling molecules that limit inflammation [130]. In our study, we observed that the gene encoding the calcium-independent phospholipase A2 VIA was strongly induced upon wounding. This phospholipase has been shown in mammals to promote adhesion, clearance of debris and ROS production to act as a chemoattractant [131]. In contrast to the work done by Stramer et al., we did not find any induction of *Drosomycin* in the wounding condition, likely due to the short time point we used. Morphological studies have shown that hemocytes modify their shape, change their adhesive properties and start to transdifferentiate into plasmatocytes and lamellocytes upon clean injury. A recent single cell analysis has deciphered some of the changes that take place in hemocyte populations 24 hours after clean injury, reflecting a change in their differentiated states [37]. Our study reveals that major transcriptome changes have already begun 45 minutes post-challenge. These changes likely reflect the transformation of resting plasmatocytes into an activated form, and their differentiation into more specialized hemocyte sub-types. The observation that the FGF ligand Pyramus that mediates blood cell progenitors differentiation in the lymph gland [50] was the most up-regulated gene 45 minutes after clean injury suggests that FGF-R pathway activation could play a prominent role in promoting the differentiation of peripheral plasmatocytes upon injury. In this work, we also show that wounding reduces apoptotic processes in blood cells while promoting cell proliferation, consistent with a previous study showing that wounding stimulates *de-novo* peripheral blood cell proliferation [132]. In accordance with these data, we observed the up-regulation of *pvf2* [102] and the PVR adaptor [133] upon clean injury. Interestingly, the PVF pathway also plays a role in hemocyte migration, as Pvf2 acts as a chemoattractant [104]. It is possible that hemocytes in close contact to the wounding site stimulate nearby hemocytes to improve wound healing and accelerate repair processes. Our results also show that plasmatocytes, like mammalian macrophages, undergo major metabolic reprogramming following injury that likely fuels their transformation into an 'activated' plasmatocyte state that is more effective at producing secreted factors or engulfing bacteria. Future studies should better characterize how immune functions are coupled with metabolic reprogramming in hemocytes.

Finally, our study reveals that hemocytes can mount specific responses to different pathogens such as *E. coli* and *S. aureus*. The early time point we chose likely prevented us from fully capturing this differentiation. The polarization of T helper cells into sub-categories in response to different cytokine environments is well established. This concept has recently been extended to innate immune cells in mammals, notably to macrophages with their M1 pro-inflammatory and M2 pro repair subtypes [134]. It is tempting to speculate that *Drosophila* plasmatocytes can be polarized toward different functions such as enhanced production of antibacterial peptides or phagocytosis according to different inflammation and metabolic states. This would explain the existence of various plasmatocyte populations in different activity states [36,37,38,39]. In this vein, it would be interesting to further characterize the transcriptome of plasmatocytes in response to other challenges such as phagocytosis of apoptotic cells and yeast. Another interesting prospect is to decipher whether sessile and circulating plasmatocytes differ in their transcriptional activities. Collectively, our study and a recent single cell analysis underline the complexities of the cellular response and open the route to functional analysis.

## Material and methods

### *Drosophila* stocks and rearing

In this work, we used *w;;Hml*$^A$.*DsRed.nls* line. Animals were reared on standard fly medium comprising 6% cornmeal, 6% yeast, 0.62% agar, and 0.1% fruit juice, supplemented with 10.6g/L moldex and 4.9ml/L propionic acid. Flies are maintained at 25˚C on a 12 h light/ 12 h dark cycle. Both males and females were used for experiments.

### Microorganism culture and infection experiments

The bacterial strains used and the respective optical density (O.D.) of the pellet at 600 nm were: *Staphylococcus aureus* (O.D. 0.5) and *Escherichia coli* (O.D. 0.5). L3 wandering larvae were pricked with a tungsten needle on the dorsal side, at the origin of the two trachea, corresponding to the A7 or A8 segments. Pricked larvae were placed into a small petri dish with fresh medium and incubated at 29˚C for 45 minutes. We then dissected larvae on a glass slide in a 120 ul PBS droplet before cell sorting.

### Cell sorting procedure

Cell sorting was performed on a BD FACS Aria II (Becton Dickinson, San Jose, CA, USA) fitted with a 100 μm nozzle and with pressure set at 20 PSI. The machine temperature was lowered at 4˚C, samples were recovered in Eppendorf tubes kept on ice and immediately resuspended in TRIzol™ (#15596018, Thermo Fisher Scientific, Waltham, MA, USA). Hemocytes were selected and sorted based on DsRed fluorescence.

### RNA extraction, sequencing and analysis

For whole larva RNA extraction, 20 animals were homogenized in tubes with glass beads and lysed with the use of a PRECELLYS™ homogenizer, with 0.5 mL of TRIzol™ reagent and 0.3 mL of chloroform. For recovery of hemocytes during FACS procedure, cells were directly resuspended in the same mix of TRIzol™-chloroform. RNA was extracted following the classical phenol-chloroform RNA extraction technique.

For all samples, RNA quality was assessed on a Fragment Analyzer (Agilent Technologies, Inc., Santa Clara, CA 95051, USA). RNA-seq libraries were prepared using 73–100 ng of total RNA and the Illumina TruSeq Stranded mRNA reagents (Illumina; San Diego, California, USA) according to the supplier's instructions. Cluster generation was performed with the

resulting libraries using the Illumina TruSeq SR Cluster Kit v4 reagents and sequenced on the Illumina HiSeq 2500 using TruSeq SBS Kit v4 reagents. Sequencing data were demultiplexed using the bcl2fastq Conversion Software (v. 2.20, Illumina; San Diego, California, USA).

The quality of the resulting reads was assessed with ShortRead (v. 1.28.0) [135]. Reads were then aligned to the reference genome (*Drosophila_melanogaster* BDGP6 dna.toplevel.fa) with TopHat (v2.1.0) and Bowtie (2.2.6.0). Mapping over exon-exon junctions was permitted by supplementing annotations (*Drosophila_melanogaster* BDGP6.87 GTF). The reads acquired in this way were used to create the lists of expressed genes (cutoff 5 counts per million [CPM] in the average of all triplicates) for each respective treatment. Unwanted variation from this data was removed by using RUVSeq (3.10) by estimating the factors of unwanted variation using residuals [136]. Differential expression analysis was performed with edgeR (3.26.4) [137].

Gene Ontology (GO) analysis was performed with Gorilla (online version http://cbl-gorilla. cs.technion.ac.il/ - July and August 2019). Two unranked lists of genes were compared, where the background set of genes was all genes with expressed with minimum of 5 CPM reads from all three combined unchallenged hemocyte reads. The target set of genes was determined by the results of the differential expression analysis of the respective treatment with the following cutoffs: CPM > 5, P-value < 0.05, FC > +/-1.88. Fold changes are expressed as real values and Log2 based values.

## Supporting information

**S1 Fig. GO terms enriched in unchallenged hemocytes compared to whole larva.** (TIF)

**S2 Fig. GO terms enriched in clean injury hemocytes compared to unchallenged hemocytes.** (TIF)

**S1 Table. Selected DEGs of interest linked to extracellular matrix reorganization, extracted from the comparison of clean injury hemocytes vs. unchallenged hemocytes.** (DOCX)

**S1 File. Raw reads numbers in all samples (L3, UC, CI, *E coli* and *S aureus*).** Average of the number of mapped reads per million reads in the respective triplicate samples. GO terms were extracted from Flybase. (XLSX)

**S2 File. Differentially expressed genes in unchallenged hemocytes versus unchallenged whole larvae samples.** Results of the differential gene expression analysis between unchallenged plasmatocytes and L3 whole larvae. Cut-off values are fold change ≥ 2, logCPM > 2, P-value < 0.05. (XLSX)

**S3 File. Transmembrane proteins coding genes identified in S2 File.** Differentially expressed genes between unchallenged hemocytes and L3 whole larvae which have the GO term 'integral component of plasma membrane' or 'cell surface'. Cut-off values are fold changes ≥ 2, logCPM > 2, P-value < 0.05. Positive FC values indicate higher expression in plasmatocytes versus L3. (XLSX)

**S4 File. Secreted proteins coding genes identified in S2 File.** Differentially expressed genes between L3 whole larvae and unchallenged hemocytes that have the GO term 'extracellular space' or 'extracellular region', which identifies putative secreted proteins regardless of the

presence of a signal peptide. Cut-off values are fold changes $\geq 2$, logCPM $> 2$, P-value $< 0.05$. Positive FC values indicate higher expression in plasmatocytes versus L3.
(XLSX)

**S5 File. Differentially expressed genes in clean injury samples versus unchallenged samples.** Results of the differential gene expression analysis between unchallenged hemocytes and hemocytes from animals with clean injury. Cut-off values are fold changes $\geq 2$, logCPM $> 2.3$, P-value $< 0.05$. Positive FC values indicate higher expression in clean injury samples versus unchallenged samples.
(XLSX)

**S6 File. Differentially expressed genes in *E. coli* samples versus clean injury samples.** Results of the differential gene expression analysis between hemocytes from animals infected with *E. coli* and hemocytes from animals with clean injury. Cut-off values are fold changes $\geq 2$, logCPM $> 2.3$, P-value $< 0.05$. Positive FC values indicate higher expression in *E. coli* samples versus clean injury samples.
(XLSX)

**S7 File. Differentially expressed genes in *S. aureus* samples versus clean injury samples.** Results of the differential gene expression analysis between hemocytes from animals infected with *S. aureus* and hemocytes from animals with clean injury. Cutoff values are fold changes $\geq 2$, logCPM $> 2.3$, P-value $< 0.05$. Positive FC values indicate higher expression in *S. aureus* samples versus clean injury samples.
(XLSX)

## Acknowledgments

We thank Hannah Westlake and Samuel Rommelaere for useful comments on the manuscript. We also thank the EPFL Flow Cytometry Core Facility (FCCF) for help in cell sorting and especially Loïc Tauzin and André Mozes. We thank the Lausanne Genomic Technologies Facility (UNIL, Lausanne, Switzerland) for RNA sequencing and especially Johann Weber.

## Author Contributions

**Conceptualization:** Elodie Ramond, Jan Paul Dudzic, Bruno Lemaitre.

**Data curation:** Jan Paul Dudzic, Bruno Lemaitre.

**Formal analysis:** Elodie Ramond, Jan Paul Dudzic, Bruno Lemaitre.

**Funding acquisition:** Bruno Lemaitre.

**Investigation:** Elodie Ramond, Jan Paul Dudzic, Bruno Lemaitre.

**Methodology:** Elodie Ramond, Jan Paul Dudzic, Bruno Lemaitre.

**Project administration:** Bruno Lemaitre.

**Resources:** Bruno Lemaitre.

**Supervision:** Bruno Lemaitre.

**Validation:** Bruno Lemaitre.

**Visualization:** Elodie Ramond, Jan Paul Dudzic, Bruno Lemaitre.

**Writing – original draft:** Elodie Ramond, Jan Paul Dudzic, Bruno Lemaitre.

**Writing – review & editing:** Elodie Ramond, Jan Paul Dudzic, Bruno Lemaitre.

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
