## [Decision Letter · Decision Letter 0]

21 May 2020

PONE-D-20-12312

Comparative RNA-Seq Analyses of Drosophila Plasmatocytes Reveal Gene Specific Signatures In Response To Clean Injury And Septic Injury

PLOS ONE

Dear Bruno,

Thank you for submitting your manuscript to PLOS ONE. It has now been evaluated by three expert reviewers. They all liked the work, but suggested minor revisions to further improve it. Therefore, we invite you to submit a revised version of the manuscript that addresses the points raised by the reviewers in their reviews.

The reviewers had trouble accessing all supplemental figures. When I tried, I was only able to download three of the seven supplemental figures. I don't know the cause of this (could be a glitch in the Plos One editorial manager). Please make sure when uploading the revision that all figures upload successfully.

We would appreciate receiving your revised manuscript within two months. If you need more time, please let us know. To enhance the reproducibility of your results, we recommend that if applicable you deposit your laboratory protocols in protocols.io, where a protocol can be assigned its own identifier (DOI) such that it can be cited independently in the future. For instructions see: http://journals.plos.org/plosone/s/submission-guidelines#loc-laboratory-protocols

We look forward to receiving your revised manuscript.

Kind regards,

Andreas

***

Andreas Bergmann, Ph.D.

Academic Editor

PLOS ONE

Journal Requirements:

2. We note that you are reporting an analysis of a microarray, next-generation sequencing, or deep sequencing data set. PLOS requires that authors comply with field-specific standards for preparation, recording, and deposition of data in repositories appropriate to their field. Please upload these data to a stable, public repository (such as ArrayExpress, Gene Expression Omnibus (GEO), DNA Data Bank of Japan (DDBJ), NCBI GenBank, NCBI Sequence Read Archive, or EMBL Nucleotide Sequence Database (ENA)). In your revised cover letter, please provide the relevant accession numbers that may be used to access these data. For a full list of recommended repositories, see http://journals.plos.org/plosone/s/data-availability#loc-omics or http://journals.plos.org/plosone/s/data-availability#loc-sequencing.

Reviewers' comments:

Reviewer's Responses to Questions

**Comments to the Author**

1. Is the manuscript technically sound, and do the data support the conclusions?

Reviewer #1: Yes

Reviewer #2: Yes

Reviewer #3: Yes

2. Has the statistical analysis been performed appropriately and rigorously? 

Reviewer #1: Yes

Reviewer #2: Yes

Reviewer #3: Yes

3. Have the authors made all data underlying the findings in their manuscript fully available?

Reviewer #1: Yes

Reviewer #2: Yes

Reviewer #3: Yes

4. Is the manuscript presented in an intelligible fashion and written in standard English?

Reviewer #1: Yes

Reviewer #2: Yes

Reviewer #3: Yes

5. Review Comments to the Author

Reviewer #1: The experimental set-up is simple, straight forward and strategic. All possible combinations were investigated and the findings well discussed in the light of the literature. The datasets will provide a basis for further studies on possible differential interaction among hemocyte (plasmatocyte) subsets in the immune response to S. aureus and E. coli.

The studies comprehensive describe the functions of the differentially expressed genes and activation networks.

The figures are of good quality and informative.

The differences in gene expression pattern in the response to S. aureus and E. coli could be elaborated a bit more. Maybe a figure (direct comparison) could help to make the findings and conclusions more transparent.

Reviewer #2: In this study, Lemaitre and colleagues have analyzed plasmatocyte-specific transcriptome profiles of Hml-positive plasmatocytes in the wild type, after clean injury, or septic injuries with Gram(-) or Gram(+) bacterium. This study is especially interesting in that immediate early transcriptional changes upon injury or infection, which do not accompany the hemocyte differentiation and are important for initiating immune reactions, are profiled for the first time. Though single-cell transcriptome analyses recently performed by other groups show hemocyte-specific RNA expressions at a single cell level, this study supplements additional insights – septic injuries in particular – into our current knowledge and itself will be a valuable resource to the community in the future.

<major points="">

1. The authors profiled Hml+ plasmatocytes by FACS sorting and compared transcriptomes of wild-type whole larvae to unchallenged hemocytes, and showed that there are hemocyte-specific gene expressions. However, some of the hemocyte-specific genes isolated in Hml+ hemocyte/whole larvae comparison could have been arisen due to the FACS sorting. To validate that FACS sorting itself does not influence the gene expression (or does exert minor alterations), although the hemocyte lineage is not changed, it will be important to add unsorted whole hemocyte transcriptome as an extra control. Several studies have released the whole RNA transcriptome of wild-type hemocytes, including the recent single-cell studies (Tattikota et al., and Cattenoz et al.,), and data are readily available for the analysis.

2. Though 45 minutes is short and may not be enough to induce differentiation of plasmatocytes, the proportion or the level of Hml may already be declining upon clean or septic injury. Is the level or the proportion of Hml+ plasmatocytes unchanged upon injury or infection? While manipulating the FACS sorting, the authors might have already noticed the information in the FACS sorting window as shown in Fig1.

<minor points="">

1. “Drosophila” needs to be italicized in line 37, 46, 73, and others.

2. line 122: 20.000, 30.000 period to a comma

3. Please match the way of writing numbers; for example, line 146: 6723, line 153: 4,477

4. add references in line 289.

5. In addition to the comparisons between EcH/clean injury and SaH/clean injury, which identify EcH- or SaH-specific gene expressions, it will be interesting to compare EcH and SaH to provide ample information on the common immune responsive genes.

6. I assume that all the analyses were done with circulating hemocytes, not the total hemocytes including sessile populations. I might have missed, but there is no information as to which population in specific was used in this study.

Reviewer #3: The manuscript by Ramond et al describes the transcriptome of Drosophila hemocytes after clean wounding and immunization with E.coli and S.aureus compared to unchallenged samples. Thus the manuscript joins a series of papers/manuscripts with a similar focus (Cho et al; Tattikota et al; Cattenoz et al), which are cited by the authors. This includes the manuscripts on BioArchives. The similarities between the studies are indicated, so altogether this is a relevant and important contribution to cellular insect immunology. One drawback of this study – admitted by the authors - is the use of only one time point for collecting hemocytes. I only wondered whether when discussing FGF signaling and Pyramus, one should refer to the Tattikota manuscript, since the authors also observe FGF signaling (breathless and branchless) as an important contribution to hemocyte activation. One important point: I could not access supplemental Figs. S4 and following and Fig. S3 does not seem to be labeled. Maybe there was a problem with downloading, although I tried different browsers. So my judgment is based on the GO enrichments.

Minor comments:

I would include the reference from the Hultmark group (Anderl et al) both when it comes to transdifferentiation/hemocyte subpopulations and perhaps some of the methodological aspects, for example they optimized FACS.

Line 80, should the second Rizki be capital?

Line 99; Fu et al. lacks year and the reference look strange in the RefList

Line 162: here I wonder about secreted immune proteins that lack a signal peptide, it is mentioned later on (when discussing Attacin-D) but since leaderless proteins are a major fraction of insect immune proteins…?

Line 264: I think lectin function during fly development is better characterized maybe one such say “immune function of lectins in flies is poorly characterized”

In Fig 2B ‘Number of elements: specific (1) or shared by 2, 3, ... lists’. The numbers are not clearly visible perhaps changing the Font color to black might make it easier to see.

  </minor></major>

6. PLOS authors have the option to publish the peer review history of their article (what does this mean?). If published, this will include your full peer review and any attached files.

Reviewer #1: No

Reviewer #2: No

Reviewer #3: No

---

## [Author Response · Author response to Decision Letter 0]

10 Jun 2020

Lausanne, June 2nd 2020,

Dear Andreas,

I hope you are doing well. 

Thank you so much for taking care of our manuscript entitled ‘Comparative RNA-Seq Analyses of Drosophila Plasmatocytes Reveal Gene Specific Signatures In Response To Clean Injury And Septic Injury (PONE-D-20-12312). We are happy to see that all the reviewers are positive about our work. We find their suggestions fair and helpful. We have addressed nearly all their comments in the revised version and hope that you will find these modifications to the manuscript satisfactory. 

We would like to thank the reviewers for their careful reading of the manuscript and their constructive suggestions. We believe that our work, although descriptive by essence, will be useful for the community in the aim of better characterizing hemocyte function. You will find below our point-by-point responses to the reviewers’ comments. We hope that this will help you in considering the manuscript for publication in PLOS ONE. 

With best regards, 

Elodie Ramond and Bruno Lemaitre

Answer to reviewers

Answers are written in green, modified and new figures are highlighted in yellow and added text in the manuscript in red.

We adapted all the references according to PLoS ONE guidelines. This led to a change in lines number. We apologize for this, and indicate the change between the old-line numbers and the new line numbers. 

Referee #1

The experimental set-up is simple, straight forward and strategic. All possible combinations were investigated and the findings well discussed in the light of the literature. The datasets will provide a basis for further studies on possible differential interaction among hemocyte (plasmatocyte) subsets in the immune response to S. aureus and E. coli. The studies comprehensive describe the functions of the differentially expressed genes and activation networks. The figures are of good quality and informative. 

We thank the reviewer for the positive appreciation of our work.

The differences in gene expression pattern in the response to S. aureus and E. coli could be elaborated a bit more. Maybe a figure (direct comparison) could help to make the findings and conclusions more transparent.

R1 / We agree with the reviewer that this part of the manuscript could be better illustrated. In the revised version, we have added a new figure (Fig 3) that highlights the similarities and differences in gene expression in these two settings. This figure is mentioned in the text at lines 412 and 424, and its description in line 427. Of note this re-analysis led us to mention in further detail the Gbp1 gene in the result section. The added text (in line 461) is “We also observed an increase in expression of Growth blocking peptide 1 (Gbp1). Gbp1 was identified as a cytokine that plays a role in IMD pathway activation upon Gram-negative bacterial challenge in the fat body and hemocytes [118]. Our study rather points to a specific induction of Gbp1 in response to a Gram-positive bacterial challenge”

Referee #2

In this study, Lemaitre and colleagues have analyzed plasmatocyte-specific transcriptome profiles of Hml-positive plasmatocytes in the wild type, after clean injury, or septic injuries with Gram(-) or Gram(+) bacterium. This study is especially interesting in that immediate early transcriptional changes upon injury or infection, which do not accompany the hemocyte differentiation and are important for initiating immune reactions, are profiled for the first time. Though single-cell transcriptome analyses recently performed by other groups show hemocyte-specific RNA expressions at a single cell level, this study supplements additional insights – septic injuries in particular – into our current knowledge and itself will be a valuable resource to the community in the future.

We are pleased by the reviewer’s positive reception of our manuscript. 

1. The authors profiled Hml+ plasmatocytes by FACS sorting and compared transcriptomes of wild-type whole larvae to unchallenged hemocytes, and showed that there are hemocyte-specific gene expressions. However, some of the hemocyte-specific genes isolated in Hml+ hemocyte/whole larvae comparison could have been arisen due to the FACS sorting. To validate that FACS sorting itself does not influence the gene expression (or does exert minor alterations), although the hemocyte lineage is not changed, it will be important to add unsorted whole hemocyte transcriptome as an extra control. Several studies have released the whole RNA transcriptome of wild-type hemocytes, including the recent single-cell studies (Tattikota et al., and Cattenoz et al.,), and data are readily available for the analysis.

R2 / We understand the concerns of the reviewer. The FACS machine was cooled down for the experiment. During the cell sorting, cells were immediately transferred in Eppendorf tubes kept on ice, and treated with Trizol within minutes. This was done to greatly minimize any further transcriptional changes after the dissection. We added in the Material and methods section, in line 614, the following sentence: « The machine temperature was lowered at 4°C, samples were recovered in Eppendorf tubes kept on ice and immediately resuspended in TRIzol™ (#15596018, Thermo Fisher Scientific, Waltham, MA, USA). » In addition, we underline in the manuscript similarities between our finding and the one reported by Irving et al., Cellular microbiology (2005) and Cattenoz et al., 2020. This makes us confident about the value of our data.

2. Though 45 minutes is short and may not be enough to induce differentiation of plasmatocytes, the proportion or the level of Hml may already be declining upon clean or septic injury. Is the level or the proportion of Hml+ plasmatocytes unchanged upon injury or infection? While manipulating the FACS sorting, the authors might have already noticed the information in the FACS sorting window as shown in Fig1.

R3 / We agree that 45 min can be considered as a short time in our model, however, we observed that larvae pricked with fluorescent bacteria carry hemocytes with internalized pathogens at 45 min. This led us to choose 45 min as a time point, to observe early transcriptional modifications. 

The proportion of hemocytes is globally very similar in our different settings, either under unchallenged, clean-injured or pricked with bacteria. We added in the Figure 2 the corresponding counts of hemocytes for each experiment. 

1. “Drosophila” needs to be italicized in line 37, 46, 73, and others.

2. line 122: 20.000, 30.000 period to a comma

3. Please match the way of writing numbers; for example, line 146: 6723, line 153: 4,477

R4 / We thank the reviewer for reading the text meticulously. We have now corrected the manuscript

4. add references in line 289.

R5 / We thank the reviewer for noticing this oversight. The line 289 corresponds now to the line 292. We have added in the manuscript a reference corresponding to the work from Geissman laboratory (ref #83).

5. In addition to the comparisons between EcH/clean injury and SaH/clean injury, which identify EcH- or SaH-specific gene expressions, it will be interesting to compare EcH and SaH to provide ample information on the common immune responsive genes.

R6 / We agree with the reviewer that a direct comparison of both gene signatures would help distinguish common and distinct responses and would clarify our purpose. For this, we added a figure (Fig 3) that defines the similarities and differences in gene expression in these two settings. This figure is mentioned in the text at lines 412 and 424, and its description in line 427.

6. I assume that all the analyses were done with circulating hemocytes, not the total hemocytes including sessile populations. I might have missed, but there is no information as to which population in specific was used in this study.

R7 / As we bled larvae without any disturbance prior dissection, the blood cell population corresponds to circulating hemocytes. We added in line 109: “The collected hemocytes thus correspond to circulating hemocytes.“

Referee #3

The manuscript by Ramond et al describes the transcriptome of Drosophila hemocytes after clean wounding and immunization with E.coli and S.aureus compared to unchallenged samples. Thus the manuscript joins a series of papers/manuscripts with a similar focus (Cho et al; Tattikota et al; Cattenoz et al), which are cited by the authors. This includes the manuscripts on BioArchives. The similarities between the studies are indicated, so altogether this is a relevant and important contribution to cellular insect immunology. 

We thank the reviewer for commenting in a positive manner our work. 

One drawback of this study – admitted by the authors - is the use of only one time point for collecting hemocytes. I only wondered whether when discussing FGF signaling and Pyramus, one should refer to the Tattikota manuscript, since the authors also observe FGF signaling (breathless and branchless) as an important contribution to hemocyte activation. 

R8 / As suggested by the reviewer, we have now mentioned in the revised version the work of Tattikota et al. concerning the FGF pathway. In line 370, we added the following sentence:

“One this line, Tattikota et al. observed an enrichment of transcripts encoding the FGF ligand Branchless and receptor Breathless in crystal cells and lamellocytes subsets respectively, and revealed a role of the FGF pathway in the encapsulation of parasitoid wasps [37].” 

This, together with our study, suggests an important role of FGF pathways in hemocyte differentiation.

One important point: I could not access supplemental Figs. S4 and following and Fig. S3 does not seem to be labeled. Maybe there was a problem with downloading, although I tried different browsers. So my judgment is based on the GO enrichments.

While this was written in the text, there were no Figs S3 and S4. We apologize for this mistake.

Minor comments:

I would include the reference from the Hultmark group (Anderl et al) both when it comes to transdifferentiation/hemocyte subpopulations and perhaps some of the methodological aspects, for example they optimized FACS.

R9 / As suggested by the reviewer, we added this reference in line 73, réf #34.

Line 80, should the second Rizki be capital?

R10 / We apologize for this annotation. We changed it to a classical writing in the core text and in the reference.

Line 99; Fu et al. lacks year and the reference look strange in the RefList

R11 / We corrected this mistake, now in line 759.

Line 162: here I wonder about secreted immune proteins that lack a signal peptide, it is mentioned later on (when discussing Attacin-D) but since leaderless proteins are a major fraction of insect immune proteins…?

R12 / The discussed section by the reviewer is now in line 151. We identified the set of ‘secreted proteins’ by selecting for the GO-terms ‘extracellular region’ and ‘extracellular space’. To our knowledge these GO-terms are attributed to proteins independently of the presence of a signal peptide. For example, Prophenoloxidases which play important immune functions lack a signal peptide and are secreted via an unknown process. Prophenoloxidases are still tagged with the GO-term extracellular region and are therefore included in our analysis (PPO2, PPO3). Additionally, the mentioned AMP Attacin-D also lacks a signal peptide but was still picked up in our analysis using the GO terms ‘extracellular region’ and ‘extracellular space’. Nevertheless, this broad Flybase classification does not guarantee that AttacinD is indeed secreted.

We added in line 1136 the following sentence: « which identifies putative secreted proteins regardless of the presence of a signal peptide ».

Line 264: I think lectin function during fly development is better characterized maybe one such say “immune function of lectins in flies is poorly characterized”

R13 / Indeed, we modified our claim as suggested by the reviewer, in line 268 by writing “The immune function of lectins is poorly characterized in Drosophila”

In Fig 2B ‘Number of elements: specific (1) or shared by 2, 3, ... lists’. The numbers are not clearly visible perhaps changing the Font color to black might make it easier to see.

R14 / To address this issue, we have changed the Font color to black and increased the size letter as suggested by the reviewer. 

References mentioned in this text

37. Tattikota SG, Cho B, Liu Y, Hu Y, Barrera V, Steinbaugh MJ, et al. A single-cell survey of Drosophila blood. Elife. eLife Sciences Publications Limited; 2020;9: 597. doi:10.7554/eLife.54818

118. Tsuzuki S, Ochiai M, Matsumoto H, Kurata S, Ohnishi A, Hayakawa Y. Drosophila growth-blocking peptide-like factor mediates acute immune reactions during infectious and non-infectious stress. Sci Rep. Nature Publishing Group; 2012;2: 210–10. doi:10.1038/srep00210

---

## [Editor Report · Decision Letter 1]

12 Jun 2020

Comparative RNA-Seq Analyses of Drosophila Plasmatocytes Reveal Gene Specific Signatures In Response To Clean Injury And Septic Injury

PONE-D-20-12312R1

Dear Bruno,

We’re pleased to inform you that your manuscript has been judged scientifically suitable for publication and will be formally accepted for publication once it meets all outstanding technical requirements.

Thank you for sending your work to PLOS ONE!

Kind regards,

Andreas

Andreas Bergmann, Ph.D.

Academic Editor

PLOS ONE

---

## [Editor Report · Acceptance letter]

16 Jun 2020

PONE-D-20-12312R1 

Comparative RNA-Seq Analyses of *Drosophila* Plasmatocytes Reveal Gene Specific Signatures In Response To Clean Injury And Septic Injury 

Dear Dr. Lemaitre:

I'm pleased to inform you that your manuscript has been deemed suitable for publication in PLOS ONE. Congratulations! Your manuscript is now with our production department. 

Kind regards, 

on behalf of

Dr. Andreas Bergmann 

Academic Editor

PLOS ONE